# Challenges and Tools for In Vitro *Leishmania* Exploratory Screening in the Drug Development Process: An Updated Review

**DOI:** 10.3390/pathogens10121608

**Published:** 2021-12-10

**Authors:** Anita Cohen, Nadine Azas

**Affiliations:** IHU Méditerranée Infection, Aix Marseille University, IRD (Institut de Recherche pour le Développement), AP-HM (Assistance Publique—Hôpitaux de Marseille), SSA (Service de Santé des Armées), VITROME (Vecteurs—Infections Tropicales et Méditerranéennes), 13005 Marseille, France; nadine.azas@univ-amu.fr

**Keywords:** zoonotic visceral leishmaniasis, in vitro exploratory antileishmanial screening, cell viability, fluorescence, bioluminescence, RNA interference, CRISPR-Cas9, omics-based methods, bio- and cheminformatics

## Abstract

Leishmaniases are a group of vector-borne diseases caused by infection with the protozoan parasites *Leishmania* spp. Some of them, such as Mediterranean visceral leishmaniasis, are zoonotic diseases transmitted from vertebrate to vertebrate by a hematophagous insect, the sand fly. As there is an endemic in more than 90 countries worldwide, this complex and major health problem has different clinical forms depending on the parasite species involved, with the visceral form being the most worrying since it is fatal when left untreated. Nevertheless, currently available antileishmanial therapies are significantly limited (low efficacy, toxicity, adverse side effects, drug-resistance, length of treatment, and cost), so there is an urgent need to discover new compounds with antileishmanial activity, which are ideally inexpensive and orally administrable with few side effects and a novel mechanism of action. Therefore, various powerful approaches were recently applied in many interesting antileishmanial drug development programs. The objective of this review is to focus on the very first step in developing a potential drug and to identify the exploratory methods currently used to screen in vitro hit compounds and the challenges involved, particularly in terms of harmonizing the results of work carried out by different research teams. This review also aims to identify innovative screening tools and methods for more extensive use in the drug development process.

## 1. Introduction

The leishmaniases are a group of vector-borne diseases common to humans and certain mammals, mainly the dog, for zoonotic visceral forms. They are caused by flagellated protozoan parasites belonging to the *Leishmania* genus. At least 20 species are encountered in human pathology [1]. These parasites are transmitted through the bite of an infected hematophagous female phlebotomine sand fly.

There was an endemic in 98 countries and territories in 2020 [1], and the geographic distribution of these diseases evolved according to the movements of the insect vector driven by climatic and environmental changes (such as deforestation and urbanization) [2]. These diseases mainly affect poor people in Africa, Asia, and South America and are associated with malnutrition, population displacement, poor housing, weak immune system, and a lack of resources [1,2]. Clinically, there are three main forms of leishmaniasis [1]: (i) Cutaneous leishmaniasis, which represents the most common form; (ii) Mucocutaneous leishmaniasis; and (iii) Visceral leishmaniasis, which is potentially fatal if left untreated. Two epidemiological types of visceral leishmaniasis coexist worldwide. Firstly, the main epidemiological type around the Mediterranean basin involves *L. infantum* and is represented by zoonotic visceral leishmaniasis. This type is clinically characterized by the triad of mad fever, anemia, and splenomegaly and is transmitted from dogs to humans. Secondly, anthroponotic visceral leishmaniasis is more commonly found in India. It is characterized by additional adenopathy and cutaneous signs and is transmitted from human to human. Today, more than 1 billion people live in areas endemic for leishmaniasis and are at risk of infection. An estimated 30,000 new cases of visceral leishmaniasis occur annually, according to the WHO [3]. In recent years, a downward trend has been observed in the number of reported visceral leishmaniasis cases, notably due to the effect of the WHO’s visceral leishmaniasis elimination program [4]. In 2020, about 87% of global visceral leishmaniasis cases were reported from eight countries (Brazil, Eritrea, Ethiopia, India, Kenya, Somalia, South Sudan, and Sudan) [1].

In this context, zoonotic visceral leishmaniasis is managed by preventing and eliminating infections in dogs, the main parasite reservoir host, but the complexity of its transmission cycle involving humans, domestic animals, wildlife, and sand fly vectors must be considered. The treatment of human cases is therefore of particular interest. The currently available antileishmanial medicines [5] are mainly represented by amphotericin B (injectable and liposomal formulations), pentamidine, and pentavalent antimonials [6]. Miltefosine is the first and only oral drug approved for leishmaniasis [7]. Paromomycin, usually administered intramuscularly, was shown to be effective in Indian visceral leishmaniasis [5]. A topical formulation is also available for cutaneous leishmaniasis [5]. Azole medicines were also reported as having variable efficacy [5]. The treatment regimen should follow national and regional guidelines, if applicable [4,8]. Nevertheless, all these medicines present several limitations. First, they are often long-term treatments. Furthermore, most of these treatments require parenteral administration. Secondly, many severe adverse side effects have long been identified for these products [7,9] and, with the liposomal formulation that reduces the nephrotoxic side effects of amphotericin B, that galenic solution carries a non-negligible cost and the requirement of cold chain maintenance [7]. Thirdly, the parasite’s development of increasing resistance [7,8,9,10] contributes to the low effectiveness of these treatments. Moreover, *Leishmania* species-dependent variations were demonstrated in drug-susceptibility [11].

Therefore, several strategies to overcome antileishmanial drugs unresponsiveness should be considered [12], as well as the promotion of research programs, notably for the screening of new antileishmanial compounds. Therefore, in this paper, we focus on the very first step of the development process of a potential drug and will review existing exploratory screening methods to identify in vitro hit compounds against *Leishmania*. First, we discuss the challenges involved in implementing exploratory in vitro screening against *Leishmania*. Then we present recent technological advances enabling the development of new research tools in the drug development process.

## 2. Challenges Involved in Implementing Exploratory Screening to Identify In Vitro Hit Compounds against *Leishmania*

Exploratory pharmacological in vitro screening is a method of scientific experimentation requiring the use of technical and technological resources to study and select, in a chemical library, the active hit compounds on a biological target. The identified hit compounds are starting points for pharmacomodulation studies to design and develop potential drugs. Therefore, exploratory in vitro screening is the very first key step in the drug development process. Given this, the instrumentation used to perform such screening is crucial. Indeed, the choice of equipment can result in significant differences in costs, required handling time, and quality of data (reproducibility and repeatability). From manual testing to semi-automated or even fully automated testing, these criteria could vary widely and represent a challenge for harmonizing results obtained by different research teams. Nevertheless, more and more examples of efficient high throughput screening against *Leishmania* are described in the literature [13,14,15,16,17,18]. 

Another challenge posed by these exploratory in vitro screening tests is the biological target itself: the *Leishmania* parasite. Indeed, the exploratory screening for *Leishmania* involves in vitro exploratory screening on a whole protozoan parasite that exists in two morphological forms (promastigote and amastigote), which means there is a multitude of potential molecular targets. The *Leishmania*’s lifecycle requires the presence of two entities: the sand fly vector and a mammalian host. Various developmental stages throughout this lifecycle are required, but these different stages (promastigote and amastigote forms) involve many variations in diverse metabolic, biochemical, and biological pathways, which were progressively detailed in the literature [19,20,21,22,23,24]. Various *Leishmania* forms are used experimentally in vitro to develop exploratory screening tests, which are promastigotes, intracellular amastigotes, and axenic amastigotes. These latter forms represent amastigotes that were adapted to grow and develop outside their host cells in a growth medium that mimics the intracellular conditions [25,26,27]. Therefore, it appears complicated to consider only one parasite form in the framework of an exploratory in vitro screening. If rapid primary screening can be performed on extracellular promastigotes and axenic amastigotes, it would be essential to confirm and identify false positives during a secondary screening of identified hit compounds on clinically relevant intramacrophagic amastigote forms [14]. Nevertheless, this strategy does not prevent false negatives represented by hit compounds specific to intracellular amastigotes without highlighted activity on extracellular forms [28]. Thus, exploratory screening tests designed to facilitate the rapid testing of a large number of compounds are usually performed on the extracellular promastigotes or axenic amastigotes, which both enable the performance of high throughput screenings with high reproducibility [29]. Although some consider that promastigotes may not be as relevant as axenic amastigotes for screening purposes [30], there is still a lack of correlation between axenic artificial forms and intracellular amastigotes [14]. As such, models using host cells currently remain the gold standard in determining compound sensitivity [31] since they provide essential information about the tested compound’s activity in the parasite’s natural environment. Nevertheless, these models also reveal variation factors and potential biases, such as a low replication rate of amastigotes compared to promastigotes [32,33], the influence of the macrophage infection rate [34], and the variety of host cells used (primary cells or cell lines) [35]. Thus, although many studies show a correlation between the results obtained on in vitro promastigotes and (axenic) amastigotes [36,37,38,39,40,41], it seems important to take all these various factors and potential biases into account before implementing secondary screenings and interpreting them.

Another challenge in terms of harmonizing work carried out by different research teams is the wide variety of existing options for detection, acquisition, and data processing systems [42]. Thus, cell viability detection was extensively used in the exploratory screening of antileishmanial compounds, especially for primary screenings on extracellular forms [27,36,37,38,39,40,41,42,43,44,45,46,47,48,49,50,51,52,53,54,55,56,57,58,59,60,61,62,63,64,65,66,67,68,69,70]. Indeed, there are many colorimetric assays that are usable, simple, inexpensive, and suitable for large-scale screening [43,44]. Some of those are poorly illustrated in the existing literature, such as the trypanothione reductase assay [45] or acid phosphatase assay [46,47,48,49], while others are widely proven, notably, those related to intracellular metabolizing salts such as the most famous MTT assay (3-(4,5-dimethylthiazol-2-yl-2,5-diphenyl tetrazolium bromide) [38,50,51,52,53,54,55,56,57,58,59]. This yellow salt is reduced to purple formazan crystals in living cells, allowing for easy determination of parasite viability. Other analogous tests are also described, such as the one using Alamar blue (resazurin) [27,60,61,62,63,64,65,66,67], an oxidation-reduction indicator that changes its color from blue to red in living cells. The use of an analog of MTT, the MTS (3-(4,5-dimethylthiazol-2-yl-5(3-carboxymethylphenyl)-2-(4-sulfophenyl)-2*H*-tetrazolium) [68,69,70], is also described. However, these tests are not very sensitive and are mainly used for low throughput screenings. Furthermore, direct counting could also be used to evaluate the leishmanicidal activity of tested compounds both in promastigote (motility of promastigotes and examination of non-viable parasites after staining) [71] and in amastigote (microscopic counting of infected macrophages and the number of parasites per macrophage after staining) assays [71]. This method has the advantage of not requiring expensive equipment, but it is time-consuming, laborious to perform, unsuitable for large-scale screening, and suffers from a lack of reproducibility. Moreover, the determination of inhibitory concentration 50% (IC_50_) may be inaccurate since the determination of parasite viability through a staining procedure is obviously difficult [25]. Different tools were developed to automate this tiresome microscopic counting [72,73]. As an example, can be cited an automated microscopic image analysis, which can be applied to the quantification of drug activity [74]. Of note is a methodology using a colorimetric β-lactamase assay described on intramacrophagic amastigotes of *L. donovani* [16]. Nevertheless, in the field of cell viability analysis, flow cytometry constitutes an interesting alternative [75,76], which is accurate and largely used to automate the reading of results. Another approach uses the detection and quantification of engineered cells expressing fluorescent gene reporters [77,78] such as green fluorescent protein (GFP) [32,79,80,81] and bioluminescent gene reporters such as luciferase [82,83,84,85,86], or a combination of both [87,88,89,90]. These methods are proven to be more sensitive and enable faster read-outs and higher throughput [91]. Moreover, reporter proteins bear or produce an easily detectable response that can be quantified even in intracellular conditions, leading to the development of many experimental models [92]. As an example, the use of a traceable bioluminescent marker, such as NanoLuc-PEST, that correlates specifically with parasite viability could provide a more relevant in vitro assay for use in both axenic and intramacrophage amastigote models. This system was already described in *L. mexicana* [93] and could be adapted to other *Leishmania* species since the employed reporter protein expression vector (pSSU-int) was already successfully used in the main species involved in pathology [29,94,95]. Furthermore, a recent comparison of several bioluminescent reporters in a cutaneous leishmaniasis model indicated that NanoLuc-parasites, despite high bioluminescence intensity in vitro, were shown to be inadequate in discriminating between live and dead parasites in drug screening protocols. Bioluminescence detection from intracellular amastigotes expressing NanoLuc-PEST, red luciferase (RedLuc), or conventional luciferase (Luc2) proved more reliable than microscopy to determine parasite killing [96]. Nevertheless, this technology also presents several limitations, including a potential antibiotic cross-resistance conferred by induced antibiotic resistance allowing the selection of recombinant parasites and the difficulty of adapting it to *Leishmania* clinical isolates. Furthermore, these genotypic modifications of parasites could result in phenotypic consequences, such as biological transformations.

## 3. Recent Technological Advances Enabling the Development of New Research Tools in the Drug Development Process

A common drawback of all the already cited methodologies is that they represent only phenotypic screenings. The molecular target(s) of identified hit compounds remains to be determined by suitable mechanistic studies, which also represent a challenge in many cases. Indeed, it often seems complicated to identify the molecular target(s) of in vitro original hit compounds, which is likely to have an impact on the number of products in development progressing from the preclinical to the clinical stage. The main impact is the acute scarcity of new drug candidates reaching clinical trials: there are 170 registered clinical trials on leishmaniases on the ClinicalTrials.gov database, of which only 21 are ongoing. Of those, 13 are interventional drug trials involving a curative treatment, and four of which involve a product being developed alone or in combination with others [97]. This difficulty in accurately identifying the molecular target(s) of hit compounds is even more surprising as the *Leishmania* genome sequence information [98,99] was described and made available, along with significant research into the parasites’ biology [100,101]. However, classical genome manipulation methods encounter some limitations in *Leishmania*, including difficulties in achieving gene knockouts because of the parasites’ high genomic plasticity [102]. Nevertheless, recent advances in various technological areas suggest new possibilities for target-based drug discovery methods [103,104,105]. Ideally, a druggable target would be critically essential to parasites’ growth or survival (such that an incomplete knockdown would result in parasites’ death), be selective, and have a catalytic site or pocket in its structure (to optimize interactions with a small molecular inhibitor) [106,107].

Therefore, RNA interference (RNAi), defined as the mechanism through which gene-specific, double-stranded RNA (dsRNA) triggers the degradation of homologous transcripts, should be helpful [108], but this approach has proven ineffective in several *Leishmania* species [109,110] due to the absence of RNAi-related genes, such as Argonaute [111,112]. Consequently, there are only a few published applications of the RNAi approach, particularly those described in cutaneous leishmaniasis pathogens such as *L. braziliensis* [108,113] and *L. mexicana* [114,115]. In visceral leishmaniasis, this RNA interference strategy was mostly used to understand the mechanism of *L. infantum* infection better, whether it be the study of the functional role of the CC chemokine receptor 5 (CCR5) [116] or that of Wnt5a signaling [117].

However, the CRISPR-Cas system is a real step forward [118]. Initially identified as a prokaryotic defense mechanism against plasmids and virus invasions, this CRISPR-Cas9 (Clustered Regularly Interspaced Short Palindromic Repeats linked to the Cas9 endonuclease protein) system has been used for genome editing and various applications for a decade [119]. Target recognition strictly requires the presence of a short protospacer adjacent motif (PAM) flanking the target site, and the subsequent R-loop formation and strand scission are driven by complementary base pairing between the guide RNA and target DNA, Cas9–DNA interactions, and associated conformational changes [120]. With it being simple, inexpensive, and efficient, this system has found some specific applications on leishmaniases since proof-of-concept in 2015 when the first CRISPR-Cas9 mediated genome editing tools were developed and demonstrated as being effective against *L. major* [121] and *L. donovani* [122]. In *Leishmania*, CRISPR-Cas9 results in double-strand DNA breaks, which can be resolved by several mechanisms, such as homologous recombination or microhomology-mediated end joining [123] as non-homologous end joining is thought to be absent in this parasite [124]. The adaptation and optimization of this approach for *Leishmania* spp. led to genetic engineering, identification of essential genes, and characterization of potential drug targets [125,126]. In particular, the method was improved in a CRISPR-Cas9 toolkit for quick and precise gene modification by integration of donor DNA, using engineered cell lines and drug selection of mutants, which was developed in *L. major*, *L. mexicana* [127] and validated in *L. donovani* [128]. As an example, this CRISPR-Cas9 approach led to the discovery that the leishmanial eIF4E cap-binding protein (LeishIF4E-3) is essential for the completion of the parasite lifecycle since the deletion of a single allele by the CRISPR-Cas9 system alters the cell morphology and results in parasite infectivity [129]. Lately, this system was successfully used to create individual null mutants for *L. mexicana* eukaryotic protein kinases (ePK) and phosphatidylinositol 3′ kinase-related kinases (PIKK). Then, the generation of the *L. mexicana* kinome gene and a further systematic functional analysis of kinases led to identifying several pathways notably regulation of the parasite’s replication, differentiation, and responses to stress which are divergent to humans, and which potentially could be targeted in the drug development process [130].

Furthermore, advances in omics-based methods also led to the identification of parasitic targets, representing another important tool for drug discovery. These technologies are increasingly used for target discovery and validation for protozoan parasites, including *Leishmania* [131,132]. Thus, a method that has been used with great success to find new drug targets in malaria [133] is in vitro evolution and whole-genome analysis (IVIEWGA). In this case, this method is more complex because the *Leishmania* genome is larger and diploid, in contrast to that of *Plasmodium*. In this method, parasites are exposed to sublethal concentrations of hit compounds already identified in phenotypic screens. In order to identify the genetic basis of their resistance, the genomes of the resistant clones are analyzed using tiling microarrays or, more typically, using whole-genome sequencing and are compared to the sensitive parent clone. In many cases, newly emerged genomic lesions are found in genes that are predicted to encode the targets. IVIEWGA recently led to the identification of two new antileishmanial drug targets. The first is the β4 subunit of the proteasome that catalyzes protein degradation through the ubiquitin-proteasome pathway [134]. This target was identified by analyzing GNF3943; the lead compound was identified using a phenotypic screen for compounds that are broadly active against *L. donovani*, *Trypanosoma cruzi*, and *Trypanosoma brucei*. Next, 3000 compounds were synthesized, from which two were used for IVIEWGA experiments that resulted in two independent mutations in the proteasome β4 subunit. These results suggested that the proteasome, which is essential in eukaryotic cells, was the target. The second is the cyclin-dependent kinase 12 (CDK12), which is involved in the control of transcription and cell division [135]. In this work, a hit compound (DDD853651/GSK3186899) from a phenotypic screen was used for a combination of chemical proteomics and IVIEWGA experiments leading to the identification of CDK12 as a parasite target. The advantage of targets discovered through this IVIEWGA method is that they are, by default, chemically validated and druggable, though complementary studies may be needed to validate the targets and their clinical relevance further. Metabolomics can also be applied to investigate metabolic pathways and potential mechanisms of action or resistance of selected hits or drugs [136]. Thus, in the context of zoonotic visceral leishmaniasis, a multiplatform metabolomic approach was developed and described [137] to elucidate the basis of the mechanisms of action and resistance of *L. infantum* to antimonial derivatives. This work was performed through an untargeted analysis of metabolic snapshots of several parasites’ populations (treated/untreated and resistant/responders) using a multiplatform approach to determine the widest possible coverage of *L. infantum* metabolome and through a ^13^C monitoring of the origin of the highlighted alterations.

Bioinformatics and cheminformatics are also powerful tools for screening and identifying drug targets in *Leishmania*, mainly based on their divergence from the host and the essential nature of their biological function. Indeed, computational approaches have gained prominence in the last two decades as reliable approaches in the prediction of drug-like compounds with much ease [106]. This approach aims to identify structure-activity relationships (SARs) in order to design optimized compounds that could be screened in silico [138,139]. In this way, a growing body of work was published on structure-based computational approaches and pharmacophore-based virtual screenings, such as the in silico high throughput screening and the molecular docking published by Saki et al. in 2019 [140]: this work identified five potent antileishmanial ligands for lipophosphoglycan (LPS) receptor, and two for γ-glutamylcystein synthase (γ-GCS) receptor in *L. infantum* among 20,000 FDA approved drugs. In 2021, similar computational approaches notably led to identifying phytochemical inhibitors of squalene synthase [141] and selective inhibitors of dihydrofolate reductase [142] in *L. donovani*. Allosteric modulators of superoxide dismutase in *L. chagasi* were also identified [143]. However, virtual screening appears ever more efficient with the development of methodological tools [140,144], but several potential biases were reported [145]. For this reason, although this approach is very interesting, fast, and a cost-efficient alternative to high-throughput screening, the results should be tested and validated experimentally [146,147,148] afterward. Among the most recently published works are those of Khatoon et al. [148], which identified original coumarin-isatin hybrids derivatives with antileishmanial activity. From 10 synthesized derivatives, molecular docking revealed all 10 compounds could successfully fit into the binding pocket of the target (Leishmanolysin gp63) of *L. tropica*. The results obtained through dynamic studies affirmed that only three compounds (Spf-6, Spf-8, and Spf-10) displayed strong binding interactions with the gp63 target. This result was experimentally validated as only these three compounds were found to be active against both promastigote and axenic amastigote forms of *L. tropica*. Other compounds were found to be inactive. This approach was also used by Peña-Guerrero et al., who validated the Lmj_04_BRCT domain as a novel therapeutic target in *Leishmania* by a structure-based drug discovery strategy, and experimentally validated in vitro a novel inhibitor (CPE2) against *L. major*, *L. amazonensis*, and *L. infantum* (promastigotes and intracellular amastigotes) among seven virtually selected compounds [149]. 

## 4. Conclusions

Today, (multi) drug resistance is a priority public health issue. Although medicine donation programs are multiplying to support the burden of neglected tropical diseases [150] such as leishmaniases, the pharmaceutical industry seems to increasingly restrict its research and development investment to new drug candidates [151]. Alternative options consist of repurposing drugs [152,153,154] or improving existing drugs, such as the gold standard amphotericin B [155]. Nevertheless, research into new drug candidates that are active against *Leishmania* must not be excluded, especially as methodological tools are becoming increasingly effective. Indeed, recent technological advances, such as those highlighted in this paper, should lead to new insights on parasite biology, with a view to identifying and characterizing a growing number of parasitic druggable targets in order to develop more assays against specific targets and drugs with original mechanism of action. In this way, such results may gradually lead to the development of more targeted high-throughput screenings, ideally performed on clinically relevant intramacrophagic amastigote forms, which could guide the development of a sustainable original drug candidate.

## Data Availability

Not applicable.

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
