# Peer review of "Challenges and Tools for In Vitro Leishmania Exploratory Screening in the Drug Development Process: An Updated Review"

_pathogens, 2021, doi:10.3390/pathogens10121608_

Round 1

Reviewer 1 Report

The manuscript deals with an extremely interesting topic and a review on tools for in vitro Leishmania screening and drug development method is necessary to be included in the scientific framework. The manuscript, despite the important message, has many scientific gaps. Many paragraphs are unclear both in the concept and in the use of the English language. Therefore, I suggest to the authors to reformulate the paper.

Reviewer 2 Report

In this article the authors try to evaluate the methods which are currently used for in vitro screening of drugs against leishmaniasis and to identify new innovative screening tools that can be used for the development of new drug targets to treat leishmaniasis. In this article major corrections must be made as it is not well structured and the sentences some times are not comprehensible. Moreover, the evaluation of the method used for screening is not well-justified and in some cases there are not the appropriate references from the bibliography to support the arguments made in the text. The article must be enriched and reshaped in order to be more comprehensible and useful for the readers of the journal. 

Introduction:

Line 28: The sentence "...by protozoan  parasites from more than Leishmania species..."  must be rephrased.

Line 29:  "Transmission to humans is..." . Maybe the authors mean ""Transmission to humans is made"? 

Lines 27-31. There is no reference

Line 34. The ... must be deleted

Line 39-41: Must be reprhased and divided in two sentenses. It does not make sence otherwise

Lines 34-48: there are no references

Lines 56-59:  there are no references

Lines 60-62: Must be reprhased and divided in two sentenses. It does not make sence otherwise

Lines 62-66: The sentences must be rephrased, they are not comprehensible and there is no flow in the information. Also references are missing

Line 68-69: "...leading to potential variation in pharmacokinetics and in drug-host immune system interaction." Not sure what the authors try to say

Line 74 and throught the text: when refering to testing against Leishmania, the authors must specify "drug screening"

Components necessary to perform an in vitro screening against Leishmania:

This paragraph does not adress the title above and it is very confusing while the evaluation of the methods in this paragraph is scarced and without any arguements as well as reference to support such evaluation. A lot of references concerning drug discovery and drug target identification for Leishmania infection are eliminated from the article. A lot of significant works have been conducted, used promastigote and intracellular amastigote forms for drug screenings. Those works have identified multiple kinases (MAPK, CDKs, GSK-3, CKs etc), phosphatases and other proteins as drug targets and proposed potential drugs to target them. This paragraph seems disoriented and has to be changed and enriched.

Lines 81-84. The sentence in not well understood and must be rephrased. 

Lines 85-87: Must be reprhased and divided in two sentenses. Otherwise, It does not make sence. Also, delete the "..."

Lines 101-104: Not sure what the authors want to say

Lines 107-109: Must be rephrased, "while" and "however"  can not be used together. 

Line 116-118: Again the sentence is not well comprehended and also is not supported by data, as many references point the exact opposite arguement. Please adress the bibliography to support the arguements in the text but also adress the existing researches.

Lines 149 and 151. Both sentences start with moreover

Lines 122-149: More references regarding each method must be added.

Recent technological advances allowing to develop new research tools

Line 163: Add the word "that" after "is". For instance, "....cited methodologies is that they represent..."

Line 164: "...and remain the difficulties in identifying the molecular targets of compounds that show activity" must be rephrased

Line 169-171: The sentence must be rephrased to be comprehensible

Line 192-195. The sentences must be rephrased to be comprehensible

Line 188-204: The work of Baker et al must be added (Systematic functional analysis of Leishmania protein kinases identifies regulators of differentiation or survival-Nature Communications, 2021)

Line 223-225. "...determined from GNF3943, hit compound identified in a phenotypic screen." must be rephrased

Line 246-248. More references with SARs studies must be added. Several studies have also combined SARs study with in vitro drug screening to identify potential drugs against Leishmania. Those studies must also be added

In general, authors seem to give an extensive example from a research in some cases of the methods mentioned in the text, while they do not give such details in others. More references have to be adressed in each one method and more examples has to be given for all the methods in order for the reader to have a more complete image of what methods are used in the bibliography and what progress has been done on drug discovery against leishmaniasis  based on these methods. Lastly, more references must be added in order to create a clearer image to the reads regarding the potential of the innovative new methods. 

Conclusions

Line 275: Not sure what the authors mean with "formed on distant forms from the pathogenic one"

References

Please use italics for in vitro, in vivo, Leishmania etc

Round 2

Reviewer 1 Report

Accept in present form

Reviewer 2 Report

In the revised manuscript, the authors provided a clearer view of the topic that is adressed in this review. They answered all the comments made by the reviewer and made text modifications accordingly. The text now is well comprehensible and concise and the topic emerged will provide an interesting reading for the readers of the journal. Thus, I propose  that the revised MS is adequate for publication in Pathogens.